# Examining the Effect of a Parent-to-Parent Intervention for Low-Income Youth with Serious Emotional and Behavioral Challenges

**DOI:** 10.3390/ijerph20146435

**Published:** 2023-07-24

**Authors:** Lindsay B. Poole, John S. Carlson, Kim Batsche-McKenzie, Justin Tate, Jane Shank

**Affiliations:** 1Department of Counseling, Educational Psychology and Special Education, Michigan State University, East Lansing, MI 48823, USA; pooleli2@msu.edu; 2Bureau of Children’s Coordinated Health Policy and Support, Michigan Department of Health and Human Services, Lansing, MI 48933, USA; batsche-mckenziek@michigan.gov (K.B.-M.);; 3Association for Children’s Mental Health, Lansing, MI 48917, USA

**Keywords:** mental health, parent-to-parent, attrition, effectiveness, acceptability

## Abstract

Background: Numerous barriers to mental health utilization exist for families of children who present with serious emotional and behavioral challenges. Evidence-based practices that facilitate equitable outcomes across diverse populations are essential to identify. This study examined possible differential service outcomes in a Medicaid-funded, parent-to-parent intervention called Parent Support Partner (PSP). Method: Data from four hundred and sixty-four parents who received PSP services were evaluated for possible demographic differences in service completion. Within-group analyses were utilized for an analysis of outcomes (parent change, child functioning; treatment acceptability) within a subset (*N* = 153) of those who completed services. Results: No racial disparities were found in those who completed PSP (43%) when compared to those who did not (57%). Regression analyses uncovered significant improvements in parent competence and confidence, as well as overall child functioning (global functioning across domains such as school, home, behaviors). Consistent with identifying evidence-based practices, findings were seen consistently across the diverse sample of those who completed PSP services. Improvements in parents’ sense of competence and confidence were correlated with perceptions of treatment acceptability. Discussion: PSP is an innovative and promising intervention with demonstrated high levels of acceptability found to increase parent confidence and self-competence to advocate for treatments that can improve the mental health functioning of their child. Future investigations of factors associated with increasing PSP service completion and outcomes in larger and more diverse populations are necessary. Implications for considering and possibly adopting this evidence-informed practice within the nursing profession are provided.

## 1. Introduction

The use of evidence-based behavioral health services within the nursing profession is hindered by institutional (limited resources, access to information challenges, inadequate staffing, and limited administrative support), interdisciplinary (e.g., a lack of communication and disconnect between training and practice), and discipline-specific barriers (e.g., nurses’ knowledge of evidence-based practices) [1]. The successful application of evidence-based practices (EBPs) hinges not only on the wide-scale dissemination of the research behind these practices but also on the issues of intervention dissemination, adoption, and implementation. Interventions that can be carried out as intended to completion, combined with patient perceptions of treatment acceptability and effectiveness, are all essential when identifying EBPs. The nursing profession must embrace and utilize those healthcare practices that not only positively impact their pediatric patients but are also aligned with the values, beliefs, and culture of the system of care and the families they work [2]. A significant need exists for the identification and utilization of evidence-based services for children experiencing serious emotional and behavioral difficulties across all behavioral health professions [3]. One in ten children experiences a severe emotional disturbance (SED) involving significant challenges arising from issues associated with mood, attention, impulsivity, conduct, post-traumatic stress, or anxiety [4]. Families of youth from diverse backgrounds including those in foster care, ethnic minorities, and children living in poverty are most at-risk of experiencing mental health difficulties [5].

Early and effective intervention is vital as barriers to accessing services are likely [6]. Only 36% of youth with SED receive mental health treatment to address their needs [7]. Involving families of youth with SED in the mental health treatment decisions is critical for retention and completion [8,9] given the high rates of attrition found within this population [10]. Coordination of services can be especially challenging for families of youth with SED given the complexity of needs, often requiring engagement with multiple systems of care [11].

Families can be especially influential in impacting their child’s mental health [12]. Clinician-led programs with a strong emphasis on parent involvement tend to be the most dominant form of mental health support available including programs such as Strategies to Enhance Positive Parenting (STEPP), Psychoeducation Responsive to Families Coping with a Child with Emotional Disorders (C-PERF), as well as Cognitive-Behavioral Family Therapy, Multi-Systemic Family Therapy, and Trauma Focused-Cognitive Behavioral Therapy (TF-CBT) [13,14,15]. However, growing research suggests peer-led (e.g., parent-to-parent) models are an especially viable approach within diverse, low-income populations [15]. Providing knowledge to families via a parent with a similar lived experience can be a helpful psychoeducational intervention that parents strongly value due to their shared social identity and validation of their experiences [16]. In addition, parent-to-parent support programs can uniquely help to increase parents’ confidence in advocating for their child within a complex array of uncoordinated services within unfamiliar systems of care [14] not only for children with mental health needs but for children with chronic conditions and developmental disabilities as well [16,17]. Systematic reviews indicate that peer-led parent programs can facilitate personal growth, competence, and feelings of support that other methods cannot provide [13,15,16].

The range of offerings varies by setting (in-home, school, outpatient), modality (standard curriculum, one-on-one sessions), length of service, and core values; however, a parent’s difficulty with managing their child’s emotional and behavioral problems remain constant. Parental strain is associated with the emotional and financial burden of caring for a child with SED [18]. Cost and insurance tend to be the primary barrier to accessing evidence-based services, but a lack of knowledge on how to navigate mental health systems of care is also an issue [19].

Highlighting the demographic differences in experience with parent-to-parent programs is worth investigating further. Previous studies have found that White/Caucasian caregivers are more likely to experience higher caregiver strain, while African American/Black caregivers are more likely to report lower perceived support [20]. Families of children diagnosed with SED and a neurodevelopmental disability (NDD) are especially at risk of experiencing significant levels of parental strain [21,22]. There are unique challenges posed by parents with children with NDDs along with programming and tips for managing their well-being [23]. Often, demographic differences are examined in isolation, though there is great benefit in determining intervention effectiveness across groups. This approach can help to solidify generalizability or targeted areas of interception for specific subpopulations.

Parent-to-parent mental health treatments can help to cultivate feelings of hope, confidence, and empowerment to address their child’s challenges [24]. Those feelings can lead to increased parental involvement in mental health services, which helps to build trust with service providers [18] and minimize parental strain [25]. Table 1 provides examples of parent-to-parent interventions provided to families of youth with SED. Each of the interventions highlighted has varying levels of evidence about service completion (e.g., attrition), effectiveness, and acceptability [26]. Long-term treatment benefits have been demonstrated via within-group, waitlist control, and randomized control trials (RCTs). Other methods for determining the impact of parent-to-parent interventions entail the use of mediation analyses [27] and exploring the relationship between caregiver strain and treatment participation [25]. Assessing intervention effectiveness within the context of attrition and parent acceptability offers a unique insight for identifying the most helpful treatments that are available for whom, and under what conditions. As can be seen in Table 1, child age is the most commonly observed variable in parent-to-parent interventions, with caregiver type, parent race, and SED type varying by study. No study to date has examined each collectively [25,26,27,28,29,30,31,32].

While numerous studies have highlighted the many benefits associated with peer-delivered mental health services, including parenting needs associated with psychoeducation, parenting stress, and parent engagement, attrition rates have varied widely from 0% to 60% [25,27,30,32], or rates of completion have not been addressed [26,28]. Some studies have closely examined treatment acceptability [25,32], while others have placed more emphasis on service adherence [27]. Arguably, assessing treatment acceptability can help to gauge engagement and future use of treatment services. Thus, assessing intervention effectiveness within the context of attrition and parent acceptability offers a unique insight for identifying the most helpful treatments available.

A paucity of studies has addressed the attrition, effectiveness, and acceptability of parent-to-parent support interventions despite being offered to parents of youth with behavioral health challenges for over 30 years [33]. It is crucial to understand whether this intervention is effective, and for whom, though few studies examined demographic information [27] to better discern who is completing parent-to-parent services and who finds it to be effective and acceptable.

To address gaps identified in the literature, three research questions of the statewide, Medicaid-funded peer support service Parent Support Partner were investigated in this study:Does parent race, caregiver type, child type, or child age predict PSP service completion?Are there subpopulation (i.e., parent race, caregiver type, child age, and child type) differences in intervention outcomes (i.e., parents and children) for those who complete PSP services?Do parent ratings of treatment acceptability predict intervention effectiveness?

## 2. Methods

### 2.1. Participants

Deidentified evaluation data collected via Research and Electronic Data Capture (REDCap) from 37 community-based PSP programs across Michigan were analyzed from those enrolled in PSP services from July 2016 to December 2019. Notably, parents enrolled in PSP services are likely involved in other services that address the well-being of their child such as Wraparound services, case management, etc. Progress data is collected every six months until PSP services are terminated. Once services have ended, parents are asked to complete an exit survey and provide a reason for exiting services. Due to the robust nature of this project, participants were isolated within a larger dataset and consisted of 464 parents receiving PSP who had the data available. The following criteria (see Figure 1) were met to address Research Question 1: (a) exit survey completed, (b) exit reason indicated, (c) a minimum of eight weeks in services, and (d) with children diagnosed with SED between the ages of 7–19. A minimum of 8 weeks was determined by the agency to be an adequate dosage to see the benefits of the services, established based on previous internal evaluation data. From this group, 153 parents had the following data available to answer Research Question 2: (a) demographic data—parent race; caregiver type; and child type, and (b) pre- and post-test scores on both outcome measures for parent competence and child functioning. Of these parents, there were 136 parents who answered the two treatment acceptability questions needed for analysis for Research Question 3. The average length of services for completers of PSP services was 8.2 months (*SD* = 5.12), while noncompleters were in service for an average of 6.7 months (*SD* = 5.08).

### 2.2. Measures 

#### 2.2.1. Attrition

Exit reasons were used to determine treatment completion (*n* = 201, 43%) and noncompletion (*n* = 263, 57%) status. Two specific exit reasons were used to operationalize PSP service completion including, “met Parent Support Partner goals/outcomes” (*n* = 145, 31%) and “parent ended services prior to goal completion because the parent was satisfied with progress” (*n* = 56, 12%). Exit reasons for noncompletion included “lost contact (e.g., could not reach family, family stopped communication)” (*n* = 102, 22%), “family moved out of service area” (*n* = 25, 5%), “parent declined/no longer in services” (*n* = 13, 3%), “parent withdrew from Parent Support Partner services due to dissatisfaction with services” (*n* = 6, 1 %), “child aged out of services at 21” (*n* = 1, >1%), “child placed out of the community (e.g., residential)” (*n* = 1, >1%), and “not specified above” (*n* = 116, 25%).

#### 2.2.2. Intervention Effectiveness

The Parent Support Partner Service Outcome Tool [34] was developed by a statewide PSP steering committee whose members were experts in parent-to-parent interventions. The pre-survey was completed by parents/primary caregivers using a 5-point Likert scale (*Never True* = 0, *Rarely* True = 1, *Sometimes True* = 2, *Usually True* = 3, *Always True* = 4). The rating form includes 24 items measuring four subscales (bridging, collaboration, developing direction, and empowerment) that link to training content and service goals. Total scores range from 0 to 96. The post-survey includes an additional five items to measure parent perceptions of alliance building and parent satisfaction with services. Initial psychometric data support the reliability and validity of the PSP Outcome Tool [35]. The findings of this study solidified a three-factor structure for the constructs: bridging collaborative relationships, caregiver feelings of empowerment, and caregiver ability to navigate resources to inform service delivery.

#### 2.2.3. Child Functioning

The Child and Adolescent Functional Assessment Scale [36] (ages 5–19), assesses global daily functioning across life domains (school/work, home, community, behaviors toward others, mood/emotions, self-harm behavior, substance abuse, and thinking) by a trained professional who is familiar with the youth and family. Subscales are scored on an ordinal scale based on a list of behavioral descriptors reflecting impairment and include the following: 0 indicates *no impairment or minimal impairment*, 10 indicates *mild impairment*, 20 indicates *moderate impairment*, and 30 indicates *severe impairment*. Total scores range from 0 to 240 and higher scores reflect significantly greater functional impairment. The reliability and validity of the CAFAS are adequate with item reliability coefficients ranging between 0.68 and 0.73 with inter-rater reliability above 0.92 [37]. This measure has been culturally validated for children with serious emotional needs, and previous participants received services in a multitude of settings including statewide agencies, residential treatment, school mental health, etc. [38,39].

#### 2.2.4. Acceptability

Two questions measured how helpful a parent/primary caregiver found PSP services and how likely they were to recommend these services to others were added to the *PSP Outcome Tool* in 2017. Each was answered on a 10-point Likert scale, *not helpful* (1) to *very helpful* (10), and *not likely* (1) to *very likely* (10), respectively. A combined treatment acceptability score (0–20) was calculated with higher scores reflecting more positive feelings about PSP services. The average treatment acceptability was 19.05 (*SD* = 2.17).

### 2.3. Procedures 

Youth who present with SED consistent with the Michigan Legislature’s Mental Health Code Act 258 of 1975 criteria of “a diagnosable mental, behavioral, or emotional disorder affecting a minor that exists or has existed during the past year for a period of time sufficient to meet diagnostic criteria specified in the most recent Diagnostic and Statistical Manual of Mental Disorders published by the American Psychiatric Association and approved by the department and that has resulted in functional impairment that substantially interferes with or limits the minor’s role or functioning in family, school, or community” [40] are eligible for a range of specialty mental health services using Community Mental Health Service Providers (CMHSP)s within the state of Michiga. Families eligible for PSP services are typically identified by a community health agency or clinician. PSP service participants are paired with a professional peer parent who becomes a part of the family’s treatment planning team. PSPs meet with their families via phone or home visits (e.g., weekly, bi-weekly) depending on the family’s individualized goals. On average, families met 27 times within a nine-month period. The purpose of this individualized service is to instill confidence and competence in parents’ ability to address their child’s needs, seek support and resources, and navigate systems of care [41]. The intervention consists of developing a family-centered plan in collaboration with the parents along with establishing individualized goals. Families meet regularly with their PSP to meet these goals before services are terminated.

Families may have a host of other mental health services (e.g., trauma-focused cognitive behavioral therapy (TF-CBT)) in conjunction with PSP depending on their devised treatment plan. To become a PSP, one must be 18 years or older and a current parent or caregiver with lived experience caring for a child with serious emotional and behavioral difficulties. Once hired by a community agency, PSP training incorporates a year-long certification process involving 40 hours of training, 10 months of coaching calls, three professional development meetings, and engagement in monthly technical assistance meetings with supervisors and PSP training staff. Agency supervisors are also supported with training and periodic round table meetings [41]. The Institutional Review Board (IRB) determined that this study did not meet the criteria for human subject research given the deidentified way data were collected and extracted for analysis. 

### 2.4. Data Analytic Plan

Data were analyzed using SPSS software. Binary logistical regression was used to examine demographic differences between completers and noncompleters. Binary logistic regression was utilized in place of chi-square to provide more precise statistics (i.e., odds ratios) to predict outcomes between groups [42]. Additionally, it can account for extraneous variables when explaining associations between variables, a limitation of using Chi-square analysis. A large sample size was utilized (*N* = 437), and the observations were independent of each other and were not repeated in the analysis. The outcome variable (e.g., completers, and noncompleters) was dichotomous, and the demographic variables (e.g., caregiver types, child type, etc.) were categorical. Multicollinearity was not a concern between the demographic variables, and linear regression was evident between the independent variable and logit transformation. Based on this information, the assumptions for this parametric test were determined to be adequate.

The sample was not stratified, instead time and interactions were evaluated using two-way Mixed ANOVA to explore whether subpopulation (e.g., parent race, child type, caregiver type, etc.) mean differences existed between pre- and post-test outcomes (PSP; CAFAS change scores). Both outcome variables were continuous, and the within-group (at least two related groups) and between-group (at least two categorical variables) qualifiers were met. No significant outliers were seen in either group (i.e., within or between groups). The outcome variables were approximately normally distributed using the Shapiro–Wilk test of normality, and lastly, Mauchly’s test of sphericity was not found to be significant.

Lastly, linear regression was utilized to examine the relationship between parent perceptions of the acceptability of the intervention and the change in PSP scores as a result of the intervention. Two continuous variables (treatment acceptability and PSP scores) were used to examine the association, and scatterplots were examined to identify a linear relationship as well as outliers. Residual errors were found to be normally distributed, and independent observations as well as homoscedasticity were assessed and deemed adequate.

## 3. Results

The sample characteristics are outlined in Table 2. Parents involved in PSP services were primarily biological mothers (*n* = 326, 70%). Biological fathers, adoptive mothers/fathers, grandmothers/fathers, foster mothers/fathers, stepmothers/fathers, other relatives, or live-in partners were combined due to sample size and referred to generically as the nonbiological mother group for analysis purposes (*n* = 138, 30%). Their child’s ages ranged from 7 to 19 years old (*M* = 11.29, *SD* = 2.98). A total of 75% percent of parents were White/Caucasian (*n* = 346), 18% were Black/African American *(n* = 84), and 7% were a combination of other groups (i.e., Hispanic origin—Mexican, Puerto Rican, South American, Mixed, Other, American Indian, and Alaska Native; *n* = 34). Most of the youth were diagnosed with SED only (*n* = 402, 89%) compared to the smaller group diagnosed with comorbid SED and NDD (e.g., autism spectrum disorder, intellectual disability: *n* = 45, 11%).

The descriptive statistics and odds ratio are provided to compare subpopulation differences between completers and noncompleters, as shown in Table 3. However, no demographic differences were found between those who completed PSP services (*n* = 201, 43%) and those who did not complete (*n* = 263, 57%). Binary logistical regression resulted in an odds ratio between White, Black, and Other parents that was not statistically significant (*p* = 0.165; see Table 3). Similarly, parents of children classified with SED only completed PSP services at similar rates compared to parents of children diagnosed with comorbid SED and NDD type (*p* = 0.333). Caregiver type (biological mothers v. nonbiological mothers) also did not predict PSP service completion (*p* = 0.408). Lastly, PSP service completion differences were similar between parents of early adolescents (ages 7–13) and parents of older adolescents (ages 14–19) (*p* = 0.83). In sum, completion status was very similar across the specific subgroups investigated and no demographic subgroups dropped out or completed PSP services more frequently than others.

Two-way mixed analysis of variance group comparisons by time, subpopulation, and PSP scores are shown in Table 4. Statistically significant improvements in primary caregivers’ scores on the PSP Service Outcome Tool [34] were found for all demographic subgroups at post-test. No significant interactions were found between time and parent race, caregiver type, child type or child age. Similarly, two-way mixed analysis of variance group comparisons by time, subpopulation, and CAFAS scores are shown in Table 5. In addition to the expected direct effects of PSP services on parents, child CAFAS scores (not directly targeted in this parent-to-parent intervention) also improved over time across the demographic subgroups. In addition, a mean group difference between age groups was found. The early adolescence group (ages 7 to 13) demonstrated significantly improved CAFAS scores compared to the improvements seen for those grouped into the older adolescence group (ages 14 to 19). For child functioning, no significant interactions (see Table 5) were found between time and parent race, caregiver type, child type, or child age.

Simple regression analyses were used to test whether change in caregivers’ sense of empowerment, self-sufficiency, and feelings of competence (PSP score) and change in the child’s functioning (CAFAS score) could be explained by treatment acceptability as shown in in Table 6. The results of the regression determined that treatment acceptability scores explained a 6.6% variance in caregivers’ sense of empowerment, self-sufficiency, and feelings of competence as measured by the PSP Service Outcome Tool [34]; *F* (1, 134) = 9.61, *p* < 0.05. For every change in one unit of treatment acceptability, PSP scores increased by about 1.68 units. A statistically significant positive linear relationship (*p* < 0.05, *r* = 0.26) was found with a small effect size. Treatment acceptability ratings were not associated with changes in the child’s CAFAS scores.

## 4. Discussion and Implications

High treatment completion rates can indicate positive parental engagement, intervention feasibility, and satisfaction with behavioral health care services. Low rates can highlight unique barriers to service utilization that may be relevant to subgroups leading to health disparities. There were equal numbers of families who completed and did not complete this parent-to-parent intervention for each demographic subgroup analyzed (i.e., parent race, caregiver type, child type, and child age). This finding contrasts with prior studies indicating that families of ethnic-racial backgrounds may be less likely to complete mental health services compared to White/Caucasian families [8], suggesting that strategies for engagement retention must be addressed globally. Addressing the practical needs of families (e.g., transportation, childcare, etc.), the use of evidence-based techniques, family readiness for change, and the integration of system supports are potential starting points to address drop-out rates in mental health service delivery [43].

The benefit of examining demographic differences within an intervention is it highlights target areas for improvement. Generally, older adolescents have a higher risk of early termination and engagement to a lesser degree compared to younger adolescents [10]. Although there were fewer parents with older adolescents (ages 14–19) enrolled in the PSP service, the two groups were generally equal in terms of completion rates. Nonbiological caregivers are more likely to terminate services compared to biological parents [9]; however, the findings failed to support differences in completion rates by caregiver type. If subpopulation differences in completion rates had been found, a logical implication would be to individualize PSP services to subgroups of diverse caregivers. Instead, the study findings appear to support this parent-to-parent support program for a wide range of diverse families of children with SED.

Only 43% of parents who began PSP services completed it. Family mobility and instability (e.g., lost contact or a move outside of the service area) were largely responsible for noncompletion and unfortunately, a common challenge experienced by health providers working with lower-income populations. Less than 1% of those who did not complete the service indicated it was due to dissatisfaction with the services. The high rate of attrition (57%) from PSP services found in this study is consistent with previous studies showing noncompletion rates as high as 60% in families accessing community-based services for youth with SED [26]. Further research that employs evidence-based strategies for family engagement and retention is needed to determine improvements [44].

Limited trust in service providers, fear of accessing or navigating services, language and cultural barriers, and feelings of dismissiveness by care professionals can be barriers to service access and completion [24,45]. While these specific variables were not measured in this study, similar completion rates across demographic subgroups are an especially encouraging sign, and study findings highlight the potential generalizability of this statewide Medicaid-funded PSP service to other populations. Professional development and ongoing training opportunities using social validity measures and/or qualitative methods can help with targeted prevention [23]. Partnering with a parent with lived experience can help lower parents’ sense of strain and burden. Finding ways to ensure PSP service completion is an important target for future research [18].

Disaggregating PSP service effectiveness had not been carried out in prior peer–parent studies (see Table 1). PSP services were found to improve parental feelings of bridging, collaboration, confidence, and empowerment of a diverse group of parents of children diagnosed with SED. The results are consistent with previous studies which demonstrated a significant change in scores between pre- and post-assessment [27,31,32]. Additionally, changes in child functioning were found although the PSP intervention does not measure this outcome directly. It is worth noting that parents of children diagnosed with comorbid SED and NDD, while limited in size (*n* = 7), demonstrated a much higher improvement in the PSP Outcome Service Tool [34], on average (+14 points), and the greatest improvement (−27 points) in the CAFAS when compared to other subgroups, thus demonstrating a meaningful impact. These results contrast with previous studies [21,22] that suggested that PSP service may not be appropriate for youth presenting with both severe behavioral difficulties and developmental needs. Minimal differences in parent or child outcomes were found between subgroups (i.e., parent race, caregiver type, and child type), except for younger children who started with less impaired child functioning who demonstrated a greater improvement in parent ratings (+18 points) compared to older children (+13 points) between pre- and post-test.

Lastly, treatment acceptability was examined to determine whether it predicted changes in parents’ and primary caregivers’ sense of empowerment and self-sufficiency or reduced child functional impairment. The findings indicated that as PSP Outcome Tool [34] Total scores increased, perceptions of treatment acceptability increased as well. The high mean score (*M* = 19.05) indicates high satisfaction with the PSP service.

### Limitations

Numerous study limitations are important to consider when interpreting the study findings. Without a comparison group or randomization, other variables may better account for the changes found in this study. Additionally, only participants who were engaged in services for a minimum of eight weeks were included in the sample which can pose a bias of fully capturing overall completion status. PSP services are typically one aspect of a child’s comprehensive treatment plan, creating additional caution when interpreting study findings as other aspects of the treatment plan may account for some improvements found and were not controlled for in this study. Medicaid-funded PSP services were designed for parents of children with SED difficulties and additional research is needed to generalize results to parents of children presenting with other types of mental health needs. Lastly, the findings are limited by the use of parent and clinician ratings presenting potential rater bias. Objective data such as behavioral observations are necessary to further validate the impact of PSP services on parents and their children.

## 5. Conclusions and Future Direction

PSP services demonstrated positive changes across all demographic subgroups of parents who received the service and their children challenged by SED. Thus, this intervention may be suitable for a multitude of settings (e.g., hospitals, schools, private outpatient clinics, etc.) and professions (e.g., social workers, psychologists, counselors, etc.) that provide behavioral health services. The appropriate next research steps would be to investigate whether specific aspects within the parent-to-parent relationship are responsible for the success of the intervention across a diverse range of healthcare settings in schools, communities, and specialized psychiatric facilities. The further study of the components of the PSP relationship (e.g., trust and confidence) may contribute to higher treatment completion rates and improved PSP training practices [14,28,44]. PSP’s ability to carry out the service as intended was not evaluated and should be in future studies to further measure the mechanisms associated with treatment changes. Lastly, replication will help to investigate the generalizability of these preliminary results. As more subpopulation demographic information becomes available, the further disaggregation of parent engagement, outcomes, and acceptability adds to the confidence and stability of the results within healthcare settings within a diverse array of communities.

This unique parent-to-parent approach to meeting the behavioral healthcare needs of children may be one that can be embraced by the nursing profession given limited systemic barriers to adoption and implementation [1,2,8]. This may be especially true within the school nursing profession, where parent-to-parent services may be especially efficient to develop and implement with effectiveness due to existing parent engagement efforts [45]. The consideration, possible adoption, and eventual implementation of parent-to-parent services aligned with the unique behavioral health nursing setting (e.g., community and schools) and practices provided are recommended.

## Figures and Tables

**Figure 1 ijerph-20-06435-f001:**
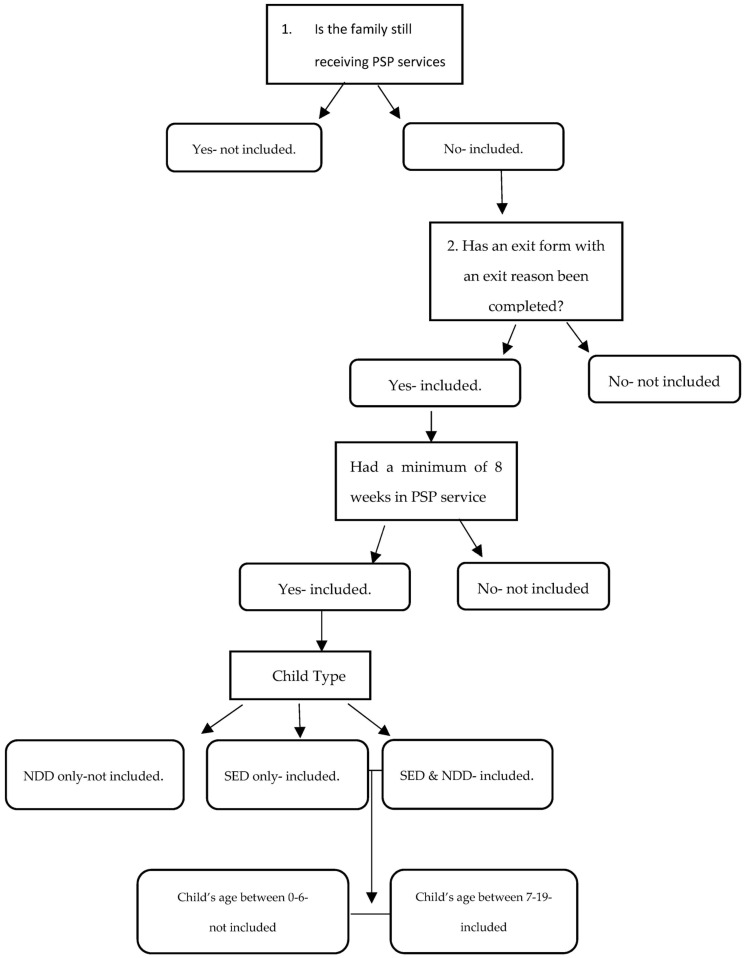
Diagram of Inclusion/Exclusion Criteria.

**Table 1 ijerph-20-06435-t001:** Summary of parent-to-parent support studies: attrition, effectiveness, and acceptability.

Parent-to-Parent Program	Demographic Characteristics Studied Comparable to Current Study	Attrition Rate	Outcome Variables	Measure of Acceptability
Parent Support Partner Program (current study)	*N* = 464 Parent race, child age, caregiver type, and SED type	56% (*n* = 263)	Parent change (bridging, collaboration, developing direction, and empowerment) Child functioning	Two questions (helpfulness and recommend to others)
Parent Connectors [25,28,29] *	*N* = 128 caregiver type and child age	0%	Caregiver strain In-school suspensions	N/A
	*N* = 348 (i.e., 180 parent connectors and 168 control parents) Parent race, caregiver type, and child age	1% (*n* = 2)	Treatment integrity	Call length, helpfulness, and recommendations to others
	*N*= 115 (i.e., 60 intervention parents and 55 controls) Caregiver type and child age	26% (*n* = 14)	Parent engagement and functioning Child impairment Academic assessment	Parent satisfaction
	*N* = 139 parent race and child age	Not Reported	Parent protective factors	N/A
Parent Partners [26] *	*N* = 2854 parent race, child age, and SED type	Not Reported	Child impairment	N/A
Parent Empowerment Program (PEP; [30])	*N* = 39 (i.e., 19 parents within the intervention groups and 20 in the control condition control groups) Child age	15% (*n* = 6)	Psychoeducation Caregiver strain Family empowermentService utilization	N/A
Smart and Secure Intervention [27] *	*N* = 15 caregiver type, child age, and parent race	60% (*n* = 9)	Child problems Parent stress Parent competence	N/A
Empowering Families, Empowering Communities [31,32]	*N* = *73* Child age	34% (*n* = 25)	Peer facilitator training Child functioning Parent stress	Peer facilitator acceptability
	*N* = 116 (i.e., 59 intervention parents and 57 waitlist parents) parent race, child age, and caregiver type	8% (*n* = 5)	Parenting behaviors Parent stress	Treatment acceptability

Note: * denotes studies that examined other factors such as feasibility and integrity.

**Table 2 ijerph-20-06435-t002:** Sample Characteristics.

Subpopulation Characteristics for This Study	*n*
Caregiver Type	
Biological mothers	326 (70%)
* Nonbiological mothers	138 (30%)
Child Age (7–19)	464 *M* = 11.29, *SD* = 2.98
Parent Race	
White/Caucasian	346 (75%)
Black/African American	84 (18%)
** Other	34 (7%)
SED Type	
SED only	402 (89%)
SED and NDD	45 (11%)

Note. * nonbiological mothers included biological fathers, adoptive mothers/fathers, grandmothers/fathers, foster mothers/fathers, stepmothers/fathers, other relatives, or live-in partners. ** The Other group consisted of Hispanic Origin—Mexican, Puerto Rican, South America; Mixed, Other, American Indian, Alaska Native.

**Table 3 ijerph-20-06435-t003:** Participant Completion Status and Logistical Regression Analyses.

Subpopulation Demographics	Completers (*n* = 201)		Noncompleters (*n* = 263)				
	Frequency	%	Frequency	%	Wald’s *c*^2^	*p*	*e* ^b^
Parent race							
White	155	77%	191	73%	1.926	0.165	1.705
Black	35	17%	49	18%	0.756	0.384	1.461
Other	11	6%	23	9%			
Caregiver type							
Biological Mother	144	72%	182	70%	0.685	0.408	1.191
Nonbiological Mother	57	28%	81	30%			
Child type (*n* = 194, 253)							
SED Only	171	88%	231	91%	0.935	0.333	0.736
SED and NDD	23	12%	22	9%			
Child Age							
Early Adolescence (ages < 13)	156	78%	199	76%	0.042	0.838	1.048
Late Adolescence (ages 14–19)	45	22%	64	24%			
Constant					1.365	0.24	0.539

Note. Binary logistical regression analysis found no statistically significant differences in completion rates.

**Table 4 ijerph-20-06435-t004:** Group Means and standard deviations and ANOVA results of PSP scores.

Subpopulation Demographics	Pre-Test		Post-test		ANOVA		
	*M*	*SD*	*M*	*SD*	Effect	*F*	*η* ^2^
Parent race (*n* = 153)							
White (*n* = 117)	73.45	1.42	82.67	1.13	T	22.73 **	0.132
Black (*n* = 25)	79.20	3.07	86.20	2.45	G	2.39	0.031
Other (*n* = 11)	76.00	4.63	89.27	3.69	TxG	0.552	0.007
Caregiver type (*n* = 153)							
Biological Mother (*n* = 94)	76.17	1.58	83.79	1.27	T	49.40 **	0.246
Nonbiological Mother (*n* = 54)	72.03	2.00	83.63	1.61	G	1.32	0.009
					TxG	2.12	0.014
Child SED type (*n* = 148)							
SED Only (*n* = 141)	75.02	1.28	83.91	1.04	T	12.92 **	0.081
SED and NDD (*n* = 7)	64.00	5.71	78.14	4.67	G	3.84	0.026
					TxG	0.673	0.005
Child Age (*n* = 153)							
Early Adolescence (*n* = 117)	74.31	1.43	83.73	1.14	T	31.42 **	0.172
Late Adolescence (*n* = 36)	75.44	2.58	83.72	2.06	G	0.792	0.000
					TxG	0.131	0.001

Note. *M* and *SD* represent mean and standard deviation, respectively, effect size = *η*^2^ or eta squared, G = group, T = time. ** *p* < 0.001.

**Table 5 ijerph-20-06435-t005:** Means and standard deviations and ANOVA results of CAFAS scores.

Subpopulation Demographics	Pre-Test		Post-Test		ANOVA		
	*M*	*SD*	*M*	*SD*	Effect	*F*	*η* ^2^
Parent race (*n* = 153)							
White (*n* = 117)	104.44	31.87	88.55	36.67	T	15.58 **	0.094
Black (*n* = 25)	106.80	36.93	82.40	26.50	G	0.172	0.002
Other (*n* = 11)	105.45	25.44	96.36	43.65	TxG	0.977	0.013
Caregiver type (*n* = 153)							
Biological Mother (*n* = 94)	102.66	32.50	84.79	34.69	T	34.92 **	0.188
Nonbiological Mother (*n* = 54)	108.47	31.55	93.39	36.68	G	2.17	0.014
					TxG	0.250	0.002
SED type (*n* = 148)							
SED Only (*n* = 141)	104.33	32.63	88.72	36.29	T	10.67 **	0.068
SED and NDD (*n* = 7)	112.86	34.50	85.71	34.57	G	0.056	0.000
					TxG	0.778	0.005
Child age (*n* = 153)							
Early Adolescence (*n* = 117)	99.23	29.91	81.28	30.97	T	23.52 **	0.135
Late Adolescence (*n* = 36)	123.33	32.68	110.28	41.09	G	25.85 **	0.146
					TxG	0.586	0.004

Note. *M* and *SD* represent mean and standard deviation, respectively, effect size = *η*^2^ or eta squared, G = group, T = time. ** *p* < 0.001.

**Table 6 ijerph-20-06435-t006:** Regression analysis summary for treatment acceptability predicting PSP and CAFAS scores.

Variable	B	*β*	*t*	*p*
Change in PSP scores				
Constant	−22.62		−2.22	0.031
Treatment acceptability	1.68	0.263	3.16	* 0.002
Change in CAFAS Scores				
Constant	−21.31		−0.87	0.386
Treatment acceptability	0.22	0.015	0.17	0.866

Note. * *p* < 0.05.

## Data Availability

Data can be requested from the corresponding author.

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
