# Peer review of "Examining the Effect of a Parent-to-Parent Intervention for Low-Income Youth with Serious Emotional and Behavioral Challenges"

_ijerph, 2023, doi:10.3390/ijerph20146435_

Round 1

Reviewer 1 Report

Thank you for the opportunity to review this manuscript. The topic is an important and understudied topic, and thus is important to add to the peer reviewed literature.

However, this study and the manner is which it is reported has many weaknesses that need to be addressed for findings to be publishable:

1)    The research questions seem relatively unrelated to each other and are not well motivated by the background section. Why is treatment completion important? This, especially, should be motivated by the background section and what reason do we have to believe that it would be different across demographic groups? Also, given the paucity of evidence about PSP – which I believe is a model that is being proliferated nationally and is even in the process of being evaluated through an NIMH funded RCT, why not just start with the basic question of “is it effective? For whom is it most effective? And Does service completion moderate effectiveness? The question of service acceptability seems most related to service completion – yet this question is unaddressed, AND the importance of service acceptability is under motivated in the background.

2)    In Table 1, Parent Connectors is listed in two different spots. I suggest combining those items and summarizing findings for each study in one row.

3)    In the methods it should be noted that there the types of services received outside of PSP are not controlled for – though this is addressed as a limitation, it should be made clear.

4)    Also, in the methods, please clarify if you are stratifying your sample and examining outcomes for each group separately, of if you are using time or intervention receipt as an interaction variable with each of the demographic variables.

5)    In the methods, you state that families had to have completed an exit interview and participated for a minimum of 8 weeks to be included in the study. This would seem to bias service completion findings and is a fundamental flaw in this study that needs to addressed.  Those that complete an exit interview are WAY more likely to have finished services, and committing 8 weeks is pretty good. What about families that drop out after 2 sessions – this is a much more likely time for drop out to occur. Perhaps talk more about the exit interview and when it is completed, and justify your eligibility criteria. Also, these should be mentioned in the limitations section.

6)    In the methods section, you discuss the demographics of the sample. This is usually reported as the first part of the findings and a sample characteristics table is presented. Adding this would strengthen the paper and moving this info the findings section is more appropriate than its current location.

7)    The PSP intervention needs to be described- as well as what makes a family eligible to receive it.

8)    What was Chi-square NOT used? I feel like you tried to address this, but it wasn’t clear. This is the much more standard way of presenting between group differences with a binary outcome?

9)    Was the completion of services determined by completing PSP or all services?

10)  NDD seems important – and I agree – but I think it needs more discussion in the background as to why it could be important and should be commented on—in the context of existing research—more in the discussion.

11)  In the discussion, the authors state “the indirect effect of improved child functioning as a result of changes in parents was also found.” I am not clear which finding you believe is indicated by this – you should explain this. Also, it would help to state at the outset (maybe in the research questions) that you are looking for an indirect effect – otherwise, this seems like a big stretch to make this claim.

Reviewer 2 Report

This submission addresses an important issue for young people and their parents.

Within its quasi-experimental  design, it demonstrates, within the clearly set out limitations of a study such as this, the value of such peer support  approaches (long known in the mental health field),  greater equality of effectiveness of such support  across different socio-economic and ethnic groups as discussed in article, which is not very often a  feature of many ‘professional’ interventions. Not essential, but I would encourage the authors to provide a little more evidence about how other programmes do not do so well on this.  

One suggestion is that the authors broaden their claim concerning  which professions (and not just nurses) these approaches may be valuable for- it would be valuable to point out it as a possibility/to be encouraged in social work also, for example.

Rather more on how peer supporters are selected and then supported would also be of value to readers.

Reviewer 3 Report

The title of the article is too long. It is recommended to limit its length so that it does not exceed 15 words maximum. Likewise, it contains various variables that can confuse the reader, so a correction and/or revision is suggested to integrate only those that are part of the purpose of the study. Check and correct.

The summary presents a functional, precise and organized structure, accounting for the various components of the study. The keywords are appropriate and consistent with the research problem.

In the introduction, they provide various theoretical and conceptual elements oriented to the contextualization of the problem, the presentation of background information and the causes associated with the central problem of the study. The use of updated bibliographical references is observed. Additionally, in table 1, they provide a series of studies that strengthen the theoretical presentation of the problem, giving an account of recent findings regarding evidence-based practices that have guided mental health interventions in children and parents with emotional and behavioral disorders. behavior. However, the section requires revision and improvement in terms of writing, grammar and punctuation. Likewise, it would be desirable for them to provide more precise scientific evidence regarding the intervention modalities reviewed (number of sessions, therapeutic approach, type of family of origin, frequency of intervention, etc.), the historical-cultural context in which they were developed, as well as as the clinical criteria used to define emotional and behavioral disorders in children.

Regarding the method, they provide basic information about the sample, the instruments and the data analysis strategy. As a suggestion, it is recommended: 1) to improve the presentation of the sample, making the inclusion and exclusion criteria explicit by means of a table or figure; 2) They must provide precise elements about the applied survey, its characteristics, structure, validation process and application; 3) the Child Functional Assessment scale does not specify whether it has suitable cultural validation in the population applied in this study. Review and substantiate.

Regarding the results, these are presented through tables and figures, under a logic from the general (descriptive statistics) to the particular (inferential statistics). As a recommendation, a synthetic, clear and precise explanation would be desirable before each table or figure, where the main findings are highlighted, using a precise style and wording without redundancies.

Regarding the discussion, it is possible to point out that the section raises a series of critical and reflective ideas based on the findings obtained, such as: 1) high treatment completion rates may reflect positive parental involvement, greater feasibility of the intervention and higher levels of satisfaction; 2) older adolescents are at higher risk of leaving an intervention early than younger ones; 3) family mobility and instability are preponderant factors in the interruption of mental health treatment; 4) limited trust in mental health providers, fear of accessing a mental health consultation, language or cultural barriers, and feelings of disdain on the part of health personnel influence access to mental health treatment in vulnerable families, among others. However, the section is superficial. Further in-depth analysis and discussion is recommended regarding the disciplinary, clinical, educational, family, social or cultural implications that could emerge from the findings for an improvement in mental health devices in vulnerable contexts, relating it to an updated review of the state of the art. . Check and correct.

The conclusions are functional and consistent with the purpose of the study. The presentation of the limitations of the study could be improved, referring to some of the comments raised in the method, in addition to explaining more clearly the need to obtain objective data for the in-depth analysis of these variables.

It is suggested to carefully review the references section, to ensure that they consistently comply with the editorial standards of the journal.

Reviewer 4 Report

As an examination of the evaluation of a parent-to-parent intervention that involved families of children who were identified as having emotional or behavioral disorders receiving Parent Support Partner services, the current study has provided key evidence of its potential.  The study examines some of the most crucial variables:  completion of the PSP services, changes in parents’ ratings of improvements in their own competence to cope, and ratings of positive changes in their children’s behavior. Especially encouraging were the favorable ratings by family members with varied racial and ethnic backgrounds. Please include your definition of child functioning in your abstract. The paper begins with a careful examination of the relevant literature, and presents the methods and results quite clearly. Please indicate who the “professional peer parents” are in Procedures.  The discussion considers the importance of the findings that provide support for the continuation and growth of this parent-centered intervention. Your consideration of the role of nursing in this intervention and of other parent-support interventions could be expanded to strengthen the connection to the major readership of the journal.  Please consider including some more recent studies in both the introduction and conclusion. Finally, the paper has a few copy editing issues.

On the whole, this article is well organized, and generally clearly written. There are a few places in the manuscript that some clarification is needed.  For example, the abstract could provide a definition of "child functioning." In the introduction, line 39 consider changing “Successful adoption…” to “Successful application…” The paper has very likely gone through several revisions, and there are some slippages requiring additional editing. For example, the results section that discusses Table 2 does not do so in the order the results appear in the table. On p. 6, the paragraph that precedes Table 2 does not do so in the order results are presented in this table. Also on p. 6, line 243 is discussing Table 3, not 4. On p. 7, the authors move on to discussing Table 4 without indicating that. On p. 8, line 304, suggest you change “empowerment to” to “empowerment of”.  Also it would be helpful to revise line 307 to read “direct effect of parent perceptions of improved child functioning.” Also p. 8, line 317, what is being reported is actually “…improvement in parent ratings (+18 points)… p 8, line 321 Should this line be changed to “PSP Outcome Total”? On p. 9 you might revise to read line 338: “received the service and had their children challenged by SED.”
